# Assessment of Cognitive Aging Using an SSVEP-Based Brain–Computer Interface System

**Saraswati Sridhar** [1,*] and **Vidya Manian** [2]

1    Southwestern Educational Society High School, Camino Pitillo, Sector Cuba, Mayagüez, PR 00682, USA
2    Dept. of Electrical and Computer Engineering, University of Puerto Rico at Mayaguez, PR-108, Mayagüez PR 00682, USA; manian@ece.uprm.edu
*    Correspondence: 21038@sesolion.com; Tel.: +1-787-617-4344

**Abstract:** Cognitive deterioration caused by illness or aging often occurs before symptoms arise, and its timely diagnosis is crucial to reducing its medical, personal, and societal impacts. Brain–computer interfaces (BCIs) stimulate and analyze key cerebral rhythms, enabling reliable cognitive assessment that can accelerate diagnosis. The BCI system presented analyzes steady-state visually evoked potentials (SSVEPs) elicited in subjects of varying age to detect cognitive aging, predict its magnitude, and identify its relationship with SSVEP features (band power and frequency detection accuracy), which were hypothesized to indicate cognitive decline due to aging. The BCI system was tested with subjects of varying age to assess its ability to detect aging-induced cognitive deterioration. Rectangular stimuli flickering at theta, alpha, and beta frequencies were presented to subjects, and frontal and occipital Electroencephalographic (EEG) responses were recorded. These were processed to calculate detection accuracy for each subject and calculate SSVEP band power. A neural network was trained using the features to predict cognitive age. The results showed potential cognitive deterioration through age-related variations in SSVEP features. Frequency detection accuracy declined after age group 20–40, and band power declined throughout all age groups. SSVEPs generated at theta and alpha frequencies, especially 7.5 Hz, were the best indicators of cognitive deterioration. Here, frequency detection accuracy consistently declined after age group 20–40 from an average of 96.64% to 69.23%. The presented system can be used as an effective diagnosis tool for age-related cognitive decline.

**Keywords:** brain–computer interface; cognitive aging; steady-state visually evoked potential; neural network; detection accuracy; band power

## 1. Introduction

Cognitive decline via deterioration of key neural networks can be caused by normal aging and/or illness (i.e., Alzheimer's disease), and often occurs before symptoms can be noted. It is well known that age significantly increases one's risk of acquiring Alzheimer's disease (AD), a severe neurodegenerative illness affecting 46.8 million people worldwide [1].

Cognitive deterioration has been explored through Electroencephalographic signaling, which enables monitoring of electrical activity in the brain with a high temporal resolution [2]. For example, Tailard et al. indicates that aging is associated with characteristic changes in EEG waveforms collected during non-REM sleep, a sleep stage with no random eye movement [2]. Furthermore, McBride et al. uses regional spectral and complexity features in EEG signals to discriminate between amnestic mild cognitive impairment (aMCI) and Alzheimer's disease (AD) [3]. Ishii et al. [4] shows that aging is characterized by significant changes in resting state oscillatory activity, event-related potentials (ERPs) elicited by cognitive tasks, functional connectivity between cerebral regions, and signal complexity.

Miraglia et al. [5] discusses the use of EEG functional network studies in order to build network topology models that could help better understand changes in brain architecture throughout an individual's lifespan. Additionally, Horvath et al. [6] examines EEG and ERP bioindicators of Alzheimer's disease, and Pagano et al. [7] examines EEG subitizing in healthy elderly subjects during working-memory and attention-related tasks.

Steady-state visually evoked potentials (SSVEPs) elicited by steadily oscillating visual stimuli are commonly employed in studies of visual perception due to their high signal-to-noise ratio (SNR) and analytical simplicity [8]. Most importantly, studies [9] have shown that SSVEP features have strong correlation with the topology of the networks they elicit. SSVEP amplitude and SNR have strong positive correlation with the efficiency and connectivity of their corresponding networks and strong negative correlation with their length, making them accurate standards of neural efficacy [10]. Such parameters affect the size of the SSVEP response generated because more efficient topological organizations of neural networks are associated with larger responses. However, few studies have focused on the effects of aging on SSVEP features; one study employs LED lights to extract Fourier amplitude and feature detection accuracy using an SSVEP-based brain–computer interface (BCI) in Amyotrophic Lateral Sclerosis (ALS) patients and subjects of varying age [11]. SSVEPs, which primarily entrain visual pathways throughout the brain, are a promising source of biomarkers of cognitive aging because the pathways stimulated by them extend throughout the entire brain. Studies examining a plethora of visual biomarkers have shown promising levels of correlation with age [12]; one prominent example is critical flicker fusion, examined by Mewborn et al., which is the frequency (flicker speed) at which the flicker of light can no longer be perceived. Critical flicker fusion, which provides insights into visual processing mechanisms, showed strong negative correlation with age [13].

SSVEP signals have a frequency range of 3.5–75 Hz; they can be categorized into particular bands, depending on their frequency. The theta, alpha, and beta bands, which are easiest to detect, are comprised of frequencies 4–8 Hz, 8–13 Hz, and 14–30 Hz, respectively. The theta band is generated in the frontal midline during deep relaxation and can be activated by rational thinking. It is also correlated with visualization or dreaming, memory, and cognitive control. The alpha band is generated in a state of relaxed alertness; its power is diminished by open eyes or increased attention levels. This rhythm, which often dominates EEG recordings, increases in prevalence and amplitude at age 7–20 and undergoes an overall decrease with age. The beta band, prevalent in the frontal lobe, is generated during a state of active concentration and is associated with problem solving, judgement, and decision-making. This band is not usually clear in EEG recordings of healthy subjects [14].

SSVEPs are commonly employed in brain–computer interfaces (BCIs), which allow direct interaction between an enhanced human brain and a computerized device without the necessity of conventional output pathways [15]. BCIs typically translate signals into meaningful commands for external devices, by restoring, at least partially, motor and communicative capabilities to individuals with compromised neural tracts. It can also facilitate interactions between humans and speech synthesizers, neural prostheses, and other assistive appliances [15]. They are also used to study different types of brain activity while the user induces a particular mental state or performs a particular task. BCIs analyze different types of EEG signals, such as P300, event-related synchronization or desynchronization (ERS/ERD), slow cortical potentials (SCPs), sensorimotor rhythms (SMR), and steady-state evoked potentials (SSEPs) [8].

The objective of this study is to develop an SSVEP-based brain–computer interface system that employs flickering light of 10 different frequencies (4, 6.6, 7.5, 8.57, 10, 12, 15, 20, 25, and 30 Hz) to collect SSVEP responses from 16 subjects spanning from age group 10–20 to >60. These responses were then epoched and analyzed to identify trends between SSVEP features and age. A predictive neural network was trained to identify level of cognitive age using these features. Section 2 presents materials and methods, Section 3 presents the results, and Section 4 presents the discussion and conclusions.

## 2. Materials and Methods

The setup consisted of five electrodes (four frontal and one occipital), positioned on a headband that the subject wore. The stimuli were presented on a laptop. Data was collected from the Cyton Biosensing Board, which received the signals from the electrodes placed on the subject's scalp in the EEG headband and wirelessly transmitted them to a USB dongle placed in a laptop computer.

### 2.1. Visual Stimulus Presentation

In this study, a single rectangular flickering stimulus (12.7 × 17.78 cm) was implemented to evoke SSVEPs in the frontal and occipital regions. This stimulus, which flickered at 4, 6.6, 7.5, 8.57, 10, 12, 15, 20, 25, and 30 Hz was programmed using MATLAB's Psychophysics Toolbox and presented on a laptop. The flickering was produced by flipping between a black screen display and the texture drawing routine of Psychtoolbox, which produced a white rectangle. The amount of flips was determined by the stimulus frequency; for example, a stimulus frequency of 4 Hz would result in four flips between the background screen and the rectangle. Each frequency was encoded with a distinct binary matrix, in which '0' encoded the black screen display and '1' encoded the white rectangle. Each stimulus frequency was presented once per subject, for 26 seconds. An intermission of 2 minutes was provided between each presentation, to minimize visual fatigue. EEG data collection was stopped 1 second after stimulus presentation.

### 2.2. Data Acquisition

EEG responses to the flickering visual stimuli were collected using OpenBCI software, via four frontal electrodes (Fp1, FpZ, Fp2, and F4), situated on a wearable headband, and one occipital electrode (Oz), arranged according to the International 10/20 system, which is one of the electrode placement systems. After the headband was fastened around the head of the subject, the occipital electrode (dry comb type) was taped (masking tape) to the back of the head and fastened under the headband. A measuring tape and marker were used to locate the electrode positions on the scalp. Then two auricular electrodes, which served as ground and reference locations, were fastened onto the subjects' ears. Conductive gel was applied to the electrodes when necessary, in order to reduce signal impedance (<100 µV). The electrode pins from the seven electrodes were connected to a Cyton Biosensing Board, which relayed the EEG signals to a USB dongle connected to a laptop computer. The USB dongle enabled the signals to be viewed and adjusted in the OpenBCI graphical user interface. The experimental setup is shown in Figure 1. Bandpass filters were applied to EEG data to filter out artifacts and noise caused by eye blinks, the presence of skin and hair, and equipment errors, among others. These filters only allowed EEG data in the range 1–50 Hz to be transmitted. The sampling rate for EEG signals collected was 250 Hz.

EEG data was collected from human subjects pertaining to age groups 10–20, 20–40, 40–60, and >60, each age group comprising four subjects, totaling 16 subjects. All subjects possessed normal or corrected-to-normal vision and if subjects had major treatments or medical issues regarding their eye health, they did not participate in this study. These subjects were covered by the Institutional Review Boards (IRB) of the University of Puerto Rico, and adequate consent and approval was obtained from all subjects. In order to control sources of variation in the data, the experiments were conducted at roughly the same time of day in a darkened room where the subjects were comfortably seated about 30.48 cm away from the computer monitor.

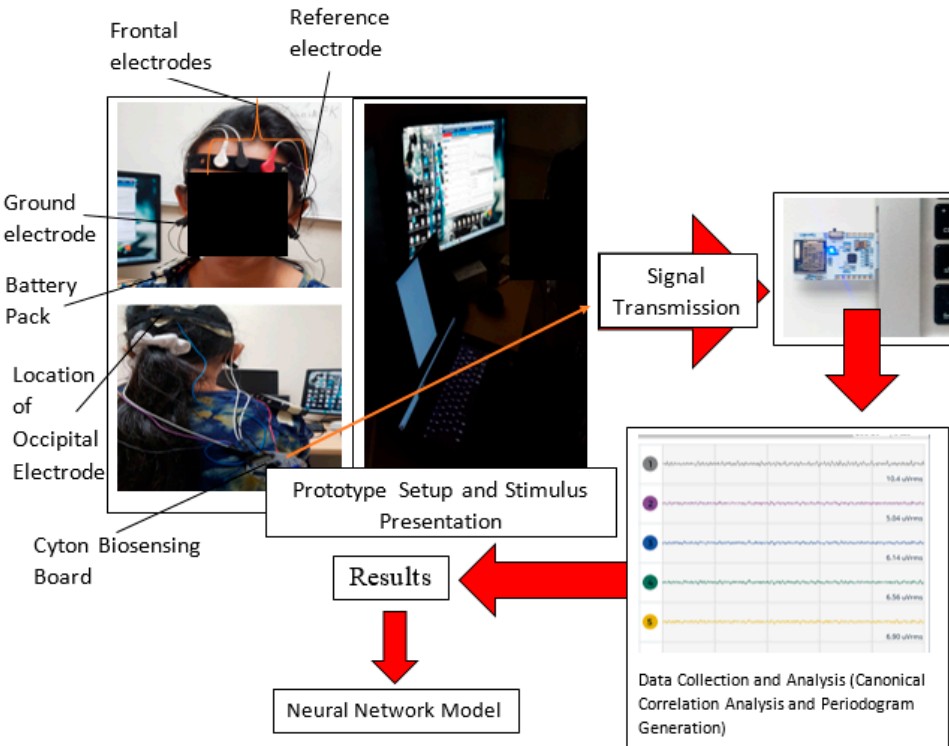

**Figure 1.** Brain–computer interface (BCI) equipment setup and experimentation.

### 2.3. Signal Processing and Feature Extraction

The EEG data, which were contained in ASCII (American Standard Code for Information Interchange) files, were converted to a MATLAB-readable format, where they were represented by a five-column matrix (five data channels) with 6540 samples or rows of points (signal amplitudes) on the EEG time-series plot (amplitude vs. time). The '.mat' file was then saved under a MATLAB variable.

Canonical correlation analysis (CCA) was used to compute a correlation coefficient between the SSVEP signals recorded at stimulus frequencies and the reference signals generated at the same frequencies [15]. Reference sinusoidal signals were generated for each stimulus frequency; each signal comprised two harmonics and was generated using the same sampling rate (250 points/sec) and number of points as the EEG signal. CCA was used to determine the reference signal that had the greatest correlation with the EEG signal; this, in turn, was used to determine which frequencies were elicited in the SSVEP. The EEG signal was processed using differing averaging intervals (2, 3, 4, and 5 s), in order to divide it into differing amounts of epochs (10, 7, 5, and 4 epochs, respectively), or trials (4160 in total). CCA was performed to determine which stimulus frequency had the greatest level of correspondence with the SSVEP in each epoch. This was used to determine intra-group detection accuracies of all 10 stimulus frequencies for each subject. This process was repeated for both the occipital (Oz) region and the frontal region, which comprised the average signal from the four frontal electrodes. This algorithm was written as a MATLAB function; each stimulus frequency is denoted by its position in a vector, and the function outputs the maximum frequency index, which is a vector displaying the frequency detected in each epoch. This information was then used to calculate the detection accuracy of theta, alpha, and beta stimulus frequencies, as the percentage of epochs where the stimulus frequency was detected correctly, in the SSVEPs.

The EEG signals were analyzed using the Fourier transform (Equation (1)), which was used to generate power spectral density (Equation (2)) plots.

$$\mathrm{X}(\omega) = \frac{1}{\sqrt{T}} \int_0^T x(t) e^{-i\omega t} dt \tag{1}$$

$$S_{xx}(\omega) = \left| X(\omega) \right|^2 \tag{2}$$

*2.4. Statistical Analysis*

After the procedures described above were completed, statistical procedures were applied to analyze and synthesize the study data. Polynomial and linear regressions were used to delineate the relationship between cognitive aging and detection accuracy of stimulus frequencies in SSVEPs, as well as SSVEP band power. The accuracy of general trends identified in this study was evaluated using analyses of variance (ANOVAs) and measures of spread, such as coefficients of variation and standard deviations.

A neural network was constructed using MATLAB's neural fitting app, in order to predict cognitive age based on frequency detection accuracy and SSVEP band power. This network consisted of 10 neurons and the input layer (training data for the model) consisted of two variables (Fourier amplitude and frequency detection accuracy), and was trained using the features extracted after epoching the data into intervals of 2, 3, 4, and 5 seconds. In this manner, there were 78 samples per subject, for 16 subjects in total. Subsequently, 70% of the data (12 samples) was used for training, 15% (two samples) for validation, and 15% for testing. The network was then trained using the Bayesian regularization algorithm, which adjusts an initial weight vector, which is used to generate predictions based on existing data, according to the input data used during training, in order to generate predictions of optimal accuracy. In this method, back-propagation occurs often to reduce prediction error. This experiment was repeated 10 times for 10-fold cross-validation, and the data were randomly divided for training, validation, and testing.

## 3. Results

Stimulus frequency detection accuracy and SSVEP Fourier amplitude as a function of age are presented below. The best cerebral regions and stimulus frequencies that were optimal in delineating cognitive aging are presented.

*3.1. Detection Accuracy of Stimulus Frequencies*

Detection accuracy of theta, alpha, and beta stimulus frequencies increased between age groups 10–20 and 20–40 and decreased continuously from age groups 20–40 to >60. This trend is shown in Figure 2 in further detail. Frequency detection accuracy is thus representative of cognitive decline only in age range 20–40 and above, because of higher levels of cognitive development. These results are shown in Table 1.

**Table 1.** Theta, alpha, and beta frequency detection accuracy in varying age groups and corresponding statistics.

| SSVEP Band (Hz) | Mean | | Standard Error | | Standard Deviation | | Coefficient of Variation | |
|---|---|---|---|---|---|---|---|---|
| Age Group | F | O | F | O | F | O | F | O |
| **Theta (4–8)** 10–20 | 90.25 | 77.75 | 2.21 | 3.57 | 5.07 | 6.24 | 6.70 | 8.60 |
| 20–40 | 93 | 92.75 | 1.22 | 3.59 | 2.22 | 5.45 | 2.40 | 6.00 |
| 40–60 | 88 | 78.5 | 3.37 | 1.50 | 3.95 | 8.42 | 4.50 | 10.6 |
| >60 | 73.75 | 76.5 | 1.89 | 3.28 | 9.75 | 7.85 | 12.7 | 10.7 |
| **Alpha (8–13)** 10–20 | 18.5 | 56 | 3.20 | 1.55 | 10.25 | 7.26 | 55.4 | 13.0 |
| 20–40 | 54.25 | 94.5 | 1.50 | 0.71 | 18.46 | 5.26 | 34.0 | 5.60 |
| 40–60 | 44.5 | 83 | 1.93 | 1.25 | 9.292 | 16.0 | 20.9 | 19.3 |
| >60 | 16.5 | 49.75 | 8.75 | 9.08 | 3.873 | 5.74 | 23.5 | 11.5 |
| **Beta (14–30)** 10–20 | 16.49 | 38.45 | 0.75 | 1.58 | 10.65 | 24.0 | 64.6 | 62.4 |
| 20–40 | 27.25 | 52 | 1.32 | 7.94 | 6.292 | 13.8 | 25.4 | 26.5 |
| 40–60 | 32 | 50.75 | 1.08 | 2.61 | 6.976 | 29.1 | 21.8 | 57.3 |
| >60 | 13.22 | 32.5 | 4.82 | 7.40 | 14.78 | 16.4 | 112 | 50.3 |

Note: SSVEP—Steady-State Visually Evoked Potential. F-Frontal. O-Occipital.

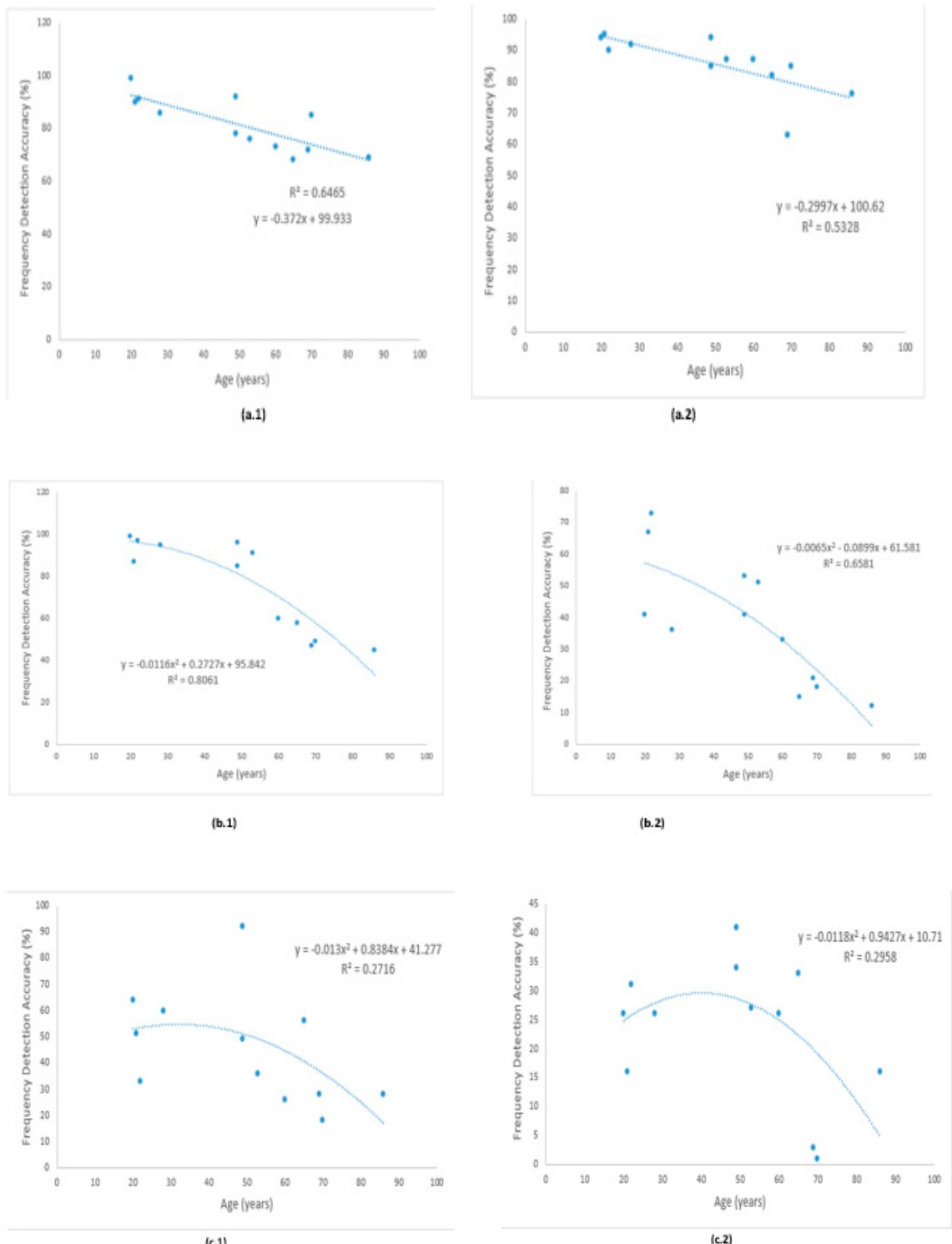

**Figure 2.** Graphical and mathematical relationship between age and detection accuracy of stimulus frequency bands in SSVEP signals: (**a.1**) Detection accuracy of theta frequency band in occipital region. (**a.2**) Detection accuracy of theta frequency band in frontal region. (**b.1**) Detection accuracy of alpha frequency band in occipital region. (**b.2**) Detection accuracy of alpha frequency band in frontal region. (**c.1**) Detection accuracy of beta frequency band in occipital region. (**c.2**) Detection accuracy of beta frequency band in frontal region.

*3.2. SSVEP Fourier Amplitude*

As demonstrated by the results, the relationship between age and SSVEP Fourier amplitude is inversely proportional, and, like frequency detection accuracy, it has a tendency to peak at age group 20–40. SSVEP Fourier amplitude as a function of age is illustrated in Table 2. Moreover, according to the data spread presented in Table 3, Fourier amplitude is the most reliable indicator of cognitive deterioration in theta and alpha SSVEPs; band power at these frequency bands as a function of age is shown in Figure 3. Fourier Amplitudes for all subjects can be found in Tables S1–S3.

**Table 2.** Band power (dB/Hz) of SSVEPs evoked by theta, alpha, and beta frequencies.

| SSVEP Band | Harmonic | Age Group (years) | | | |
| --- | --- | --- | --- | --- | --- |
| | | 10–20 | 20–40 | 40–60 | >60 |
| Theta (4–8 Hz) | 1 | 20.55 | 25.42 | 18.01 | 12.333 |
| | 2 | 19.3 | 28.717 | 17.47 | 14.174 |
| | 3 | 12.39 | 13.55 | 9.137 | 13.04 |
| | 4 | 12.18 | 16.313 | 8.369 | 9.6 |
| | Mean | 16.105 | 21 | 13.2465 | 12.28675 |
| Alpha (8–13 Hz) | 1 | 22.5 | 27.72 | 15.773 | 12.583 |
| | 2 | 14.225 | 26.71 | 18.367 | 10.816 |
| | 3 | 13.075 | 15.538 | 10.725 | 6.1018 |
| | 4 | 10.347 | 13.818 | 10.786 | 5.2189 |
| | Mean | 15.03675 | 20.9465 | 13.91275 | 8.679925 |
| Beta (14–30 Hz) | 1 | 20.953 | 23.1 | 16.835 | 13.473 |
| | 2 | 18.08 | 19.4 | 17.868 | 14.53 |
| | 3 | 10.505 | 12.123 | 9.2393 | 8.168 |
| | 4 | 9.842 | 8.386 | 6.7793 | 6.2207 |
| | Mean | 14.845 | 15.75225 | 12.6804 | 10.597925 |

**Table 3.** Statistics of spread (standard deviation and coefficient of variation) for band power of first four SSVEP harmonics evoked by theta, alpha, and beta frequencies.

| | | Harmonic | | | | | | | |
| --- | --- | --- | --- | --- | --- | --- | --- | --- | --- |
| | | 1st Harmonic | | 2nd Harmonic | | 3rd Harmonic | | 4th Harmonic | |
| SSVEP Band | Age Group | Std. Dev. | Coeff. Var. | Std. Dev. | Coeff. Var. | Std. Dev. | Coeff. Var. | Std. Dev. | Coeff. Var. |
| Theta (4–8 Hz) | 10–20 | 1.56978 | 7.64 | 0.91099 | 0.392 | 1.41 | 16.12 | 1.6217 | 7.38 |
| | 20–40 | 0.6149 | 2.42 | 2.215 | 7.71 | 4.59 | 33.86 | 3.17 | 19.44 |
| | 40–60 | 3.66 | 20.33 | 3.99 | 22.82 | 2.57 | 28.11 | 3.054 | 36.49 |
| | >60 | 2.57 | 20.81 | 5.33 | 37.58 | 3.7 | 28.37 | N/A | N/A |
| Alpha (8–13 Hz) | 10–20 | N/A | N/A | 2.14 | 15.06 | 1.62 | 12.39 | 4.47 | 43.23 |
| | 20–40 | 1.76 | 6.37 | 1.82 | 6.82 | 1.92 | 12.37 | 3.32 | 24.02 |
| | 40–60 | 3.24 | 20.56 | 5.43 | 29.54 | 6.66 | 62.09 | 2.08 | 19.27 |
| | >60 | 1.54 | 12.28 | 4.81 | 44.5 | 2.73 | 44.7 | 2.52 | 48.2 |
| Beta (14–30 Hz) | 10–20 | 4.03 | 19.25 | 6.58 | 36.41 | 3.25 | 30.95 | 4.07 | 41.37 |
| | 20–40 | 8.68 | 37.57 | 6.47 | 33.34 | 5.62 | 46.37 | 6.01 | 71.66 |
| | 40–60 | 3.96 | 23.51 | 3.37 | 18.85 | 4.72 | 51.07 | 2.54 | 37.51 |
| | >60 | 0.6025 | 4.47 | 6.12 | 42.09 | 1.38 | 16.83 | 3.07 | 49.43 |

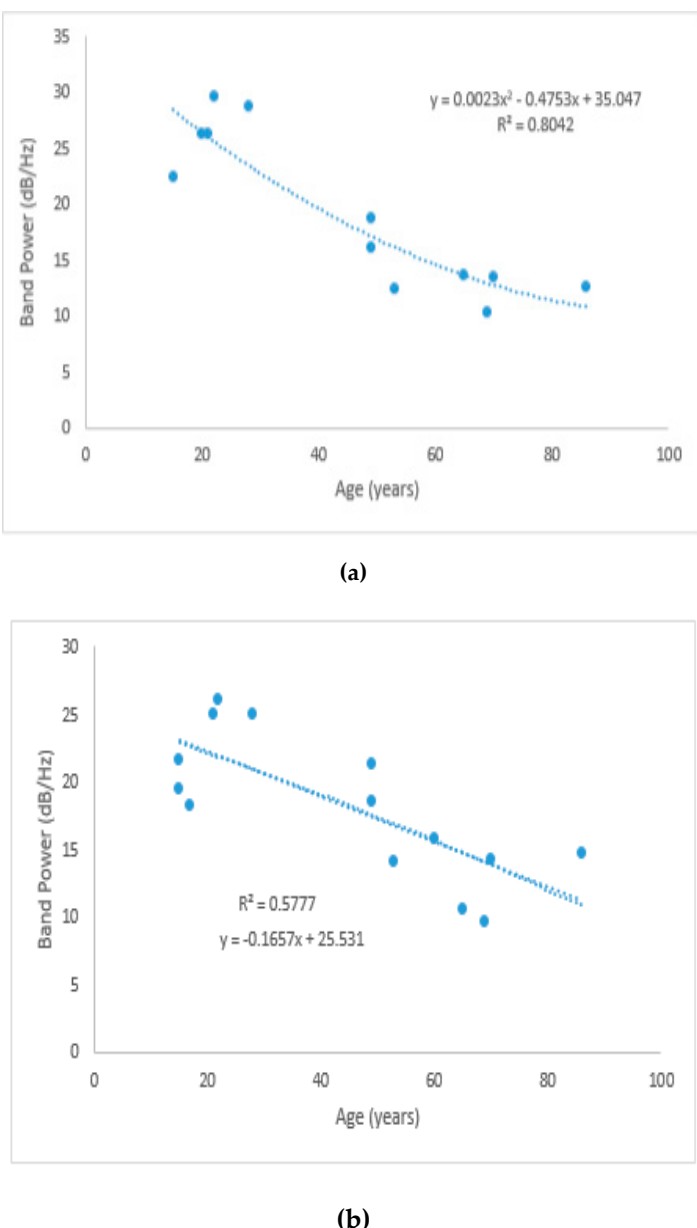

**(a)**

**(b)**

**Figure 3.** Band power (dB/Hz) of SSVEPs evoked by (**a**) theta and (**b**) alpha frequencies.

*3.3. Cognitive Function, Development, and Deterioration*

As mentioned before, SSVEP frequency detection accuracy and band power/Fourier amplitude peak between ages 20–40 (approximately 30) and decrease thereafter. Furthermore, Table 4 shows that the Pearson correlation coefficients between age, frequency detection accuracy, and SSVEP band power, which exhibit inverse variation between the two factors, are significantly stronger when excluding age group 10–20 than when including it, suggesting that the method is feasible only for age groups above 20.

**Table 4.** Coefficients of correlation with age for SSVEP band power and stimulus frequency detection accuracy including (light beige) and excluding (light blue) age group 10–20.

| | Stimulus Frequency Band (Hz) | | | | | |
|---|---|---|---|---|---|---|
| | Theta (4–8) | | Alpha (9–13) | | Beta (14–30) | |
| SSVEP Feature | F | O | F | O | F | O |
| Frequency Detection Accuracy | −0.62 | −0.29 | −0.27 | −0.34 | −0.09 | −0.21 |
| | −0.73 | −0.804 | −0.81 | −0.87 | −0.37 | −0.45 |
| Band Power | −0.80 | | −0.82 | | −0.54 | |
| | −0.87 | | −0.91 | | −0.55 | |

Note: SSVEP—Steady-State Visually Evoked Potential. F-Frontal. O-Occipital.

### 3.4. Optimal Frequency Range and Cerebral Region for Cognitive Assessment

Results showed that the alpha frequency band was the best indicator of cognitive decline. Figure 4 shows a clear correlation between frequency detection accuracy and age; as shown, the alpha stimulus frequencies elicited the greatest change in detection accuracy as a function of age. Moreover, as shown in Table 1, the variations obtained for theta and alpha frequency stay within an acceptable range (<30%), whereas spread in beta frequency detection accuracy often exceeds standards of reliability. Furthermore, although frontal responses demonstrated feasibility as indicators of cognitive aging, trends pertinent to occipital responses were significantly stronger, as demonstrated by Figure 5. Thus, as demonstrated by $R^2$ values in Figures 2 and 3, occipital responses to alpha frequencies are the best indicators of cognitive deterioration.

Analysis of variance (ANOVA) with $p < 0.05$ showed that age group, frequency band, and electrode region have substantial effect on frequency detection accuracy and SSVEP band power. Furthermore, EEG signals elicited by 7.5 Hz ($p = 0.00037$) and 12 Hz ($p = 0.0008$) were most impacted by age. The alpha stimulus frequencies were, on average, the strongest indicators of cognitive aging. This finding, however, was not present in the effect of age group on detection accuracy of beta stimulus frequencies, suggesting that the variation in beta frequency detection accuracies is too high for the data to be considered reliable.

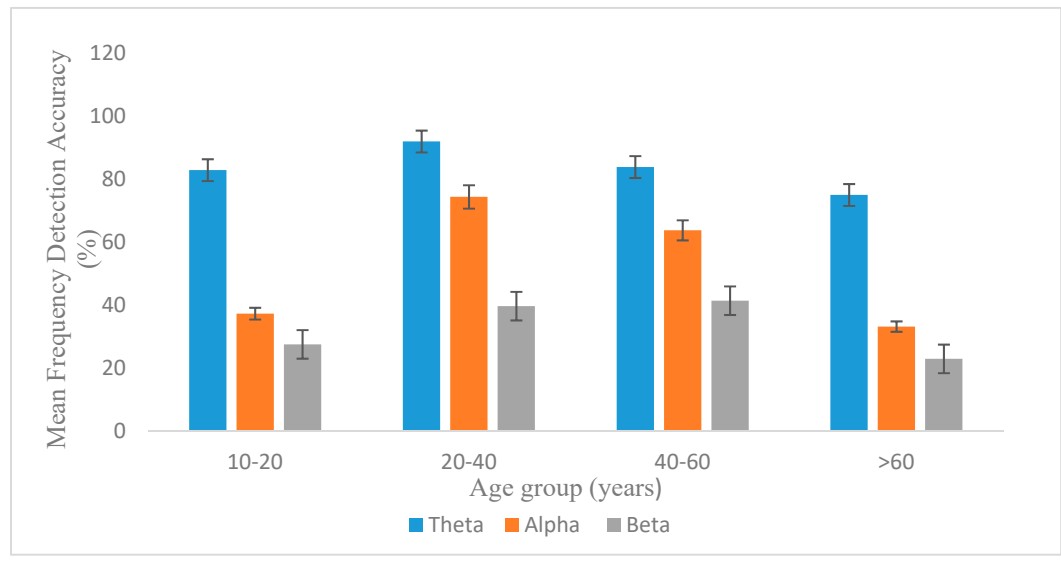

**Figure 4.** Mean detection accuracy of theta, alpha, and beta stimulus frequencies in SSVEP signals pertaining to varying age groups.

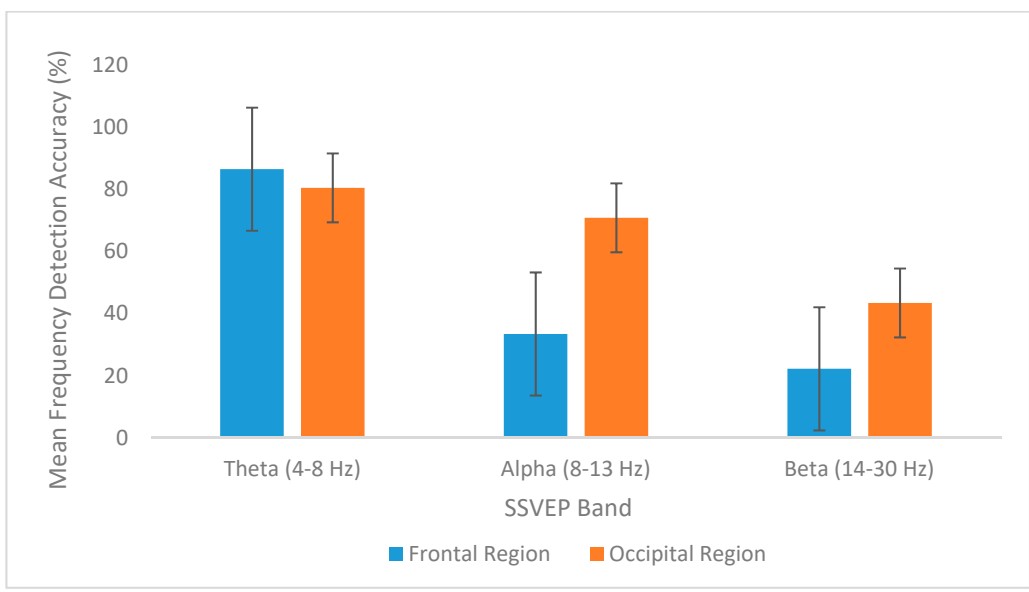

**Figure 5.** Mean detection accuracy of theta, alpha, and beta stimulus frequencies in frontal and occipital SSVEP signals.

### 3.5. Prediction using Neural Network

A neural network was trained (Bayesian regularization) using stimulus frequency detection accuracy and band power (dB/Hz) of SSVEPs evoked by alpha stimulus frequencies in age groups 20–40 and above. This neural network displayed high predictive power, as the correlation coefficient between the target values and the output values was high, ~0.988. Table 5 shows the neural network outputs when tested with random inputs. The training, testing, and validation results of the neural network are shown in Figure 6, and the training performance of the neural network at varying data segments is shown in Figure 7.

**Table 5.** Neural network predictions of cognitive age when given random frequency detection accuracies (alpha band, occipital region) and band power values as inputs.

| Inputs | | Output (Predicted Cognitive Age) |
|---|---|---|
| **Alpha Frequency Detection Accuracy (%)** | **Alpha Band Power (dB/Hz)** | |
| 94.0 | 28.1 | 22.7 |
| 86.5 | 25.9 | 20.2 |
| 59.0 | 15.8 | 49.0 |
| 45.0 | 16.1 | 53.7 |
| 57.0 | 12.8 | 68.6 |
| 46.0 | 10.3 | 81.3 |

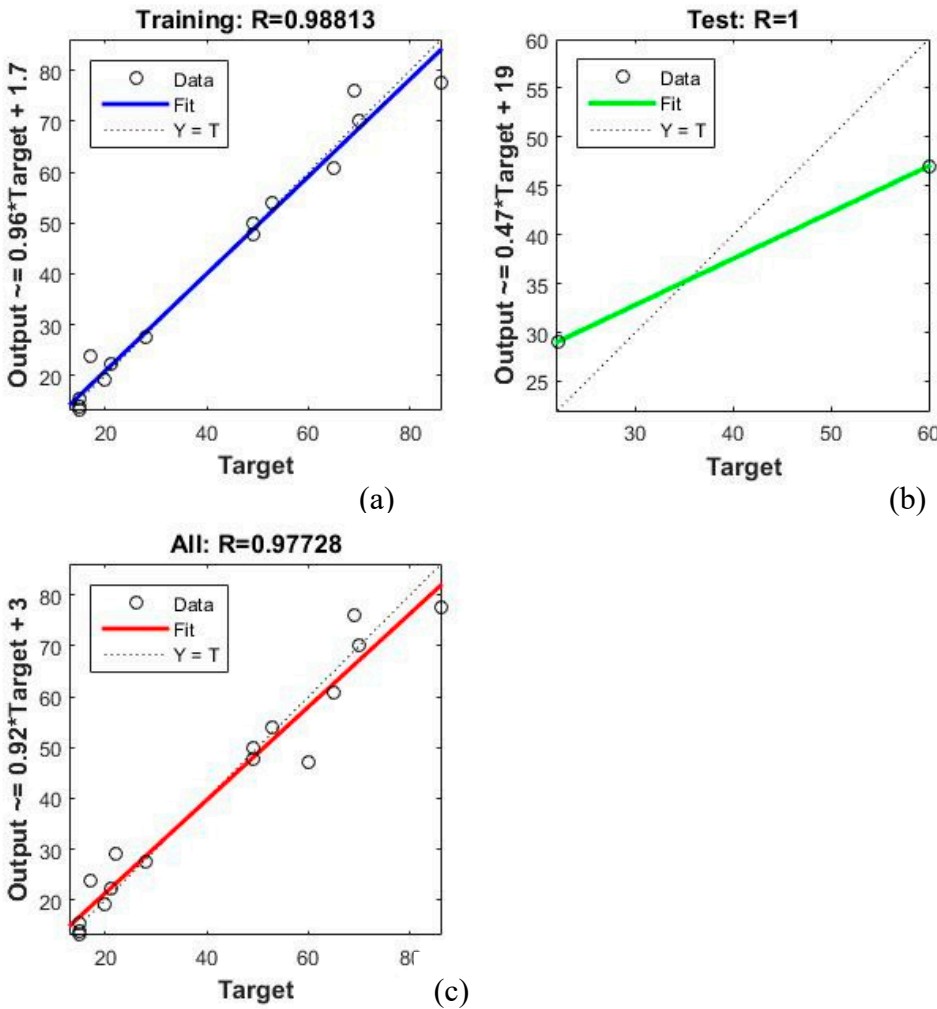

**Figure 6.** Output of the neural network model according to provided targets during (**a**) training (blue), (**b**) validation (green), and (**c**) testing (red), with corresponding correlation coefficients.

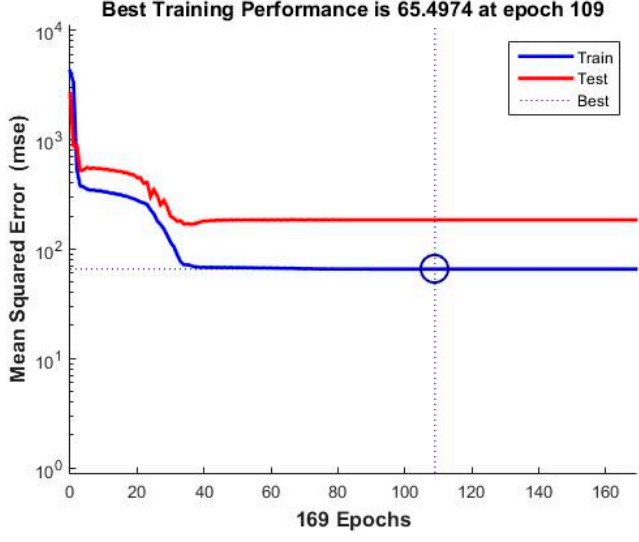

**Figure 7.** Training performance of the neural network.

## 4. Discussion and Conclusions

The SSVEP-based diagnosis BCI system was used with subjects of varying age to determine its ability to detect cognitive aging, and if possible, identify the manner in which it is manifested by features of steady-state visually evoked potentials (SSVEPs). The results of this study suggest that SSVEPs elicited by flickering stimuli may be valuable biomarkers of cognitive deterioration because SSVEP features such as band power and presence of stimulus frequencies in the signal, exhibited a sharp decline as a function of age, particularly in EEG signals elicited by alpha (8–13 Hz) flicker frequencies. These results were used to train an artificial neural network that effectively predicts cognitive age based on SSVEP band power and detection accuracy of stimulus frequencies in the signal.

The results of this study suggest that detection accuracy of stimulus frequencies in SSVEP signals indicate cognitive decline in age groups 20–40 and above. As demonstrated by Figure 4, frequency detection accuracy within the SSVEP signal reaches a peak for age group 20–40 and declines continuously afterwards. Similarly, other studies report a recession in accuracy in elderly subjects [11]. The increase in detection accuracy between age groups 10–20 and 20–40 can be attributed to ongoing cognitive development, which, according to recent studies may continue up to the mid-twenties; cognitive deterioration typically begins in the 30s or 40s [3]. For this reason, it is more practical to use this application to gauge cognitive function for this age range. This is further corroborated by Table 4, which shows that the Pearson correlation coefficients between age and frequency detection accuracy and SSVEP band power, which exhibit inverse variation between the two factors, are significantly stronger when excluding age group 10–20 than when including it.

Figure 2 demonstrates an inversely proportional relationship between age (20 and above) and detection accuracy of theta, alpha, and beta stimulus frequencies in the SSVEP signal. However, while the relationship between theta frequency detection accuracy and age seems to be linear, as shown in Figure 2a, detection accuracy of alpha and beta frequencies, shown in Figures 2b and 3c, reaches a plateau between ages 20 and 40, and exhibits a precipitous decline afterward. Likewise, studies indicate that larger SSVEP responses are associated with more efficient functional network topology in the human brain, suggesting that this trend could be caused by the aging of these systems [9]. Additionally, while detection accuracy of theta stimulus frequencies in SSVEPs can be effectively modeled using linear regression, detection accuracies of alpha and beta frequencies are better represented by quadratic regression models.

Similar trends are manifested by SSVEP band power (or Fourier amplitude) at theta, alpha, and beta frequencies, as can be seen in Figure 3 and Table 2. SSVEP band power displays an overall decrease as a function of age, in EEG responses to all three frequency bands. As in the case of frequency detection accuracy, its relationship with age can be delineated using quadratic regression, and it peaks at age group 20–40. Table 2 also demonstrates that SSVEP Fourier amplitude is typically highest at the first and second harmonics. Furthermore, Table 3 shows that the first harmonic of theta-evoked and alpha-evoked SSVEPs are the most reliable indicators of cognitive deterioration.

The results demonstrate that although these trends can be discerned in EEG responses to all frequency bands, the alpha band was shown to be the best indicator of cognitive decline. As shown in Figure 4, the alpha band displays the most change as a function of age. In addition, as shown in Figures 2 and 3, detection accuracy of and SSVEP band power at alpha stimulus frequencies display the highest $R^2$ values (0.80 occipital and 0.81, respectively), and thus demonstrate the greatest conformity with the previously described trends. The lowest $R^2$ values (0.3046 for frequency detection accuracy and 0.2923 for SSVEP band power) occurred for EEG responses to beta stimulus frequencies, shown in Figure 2c, suggesting that these frequencies are the least reliable indicators of cognitive decline. Furthermore, in Figure 2 (c.2), various outliers can be noted. These outliers may arise as a result of high levels of variation, which are typical of human systems, within EEG responses to beta stimuli. It is interesting to note that the highest outlier, occurring between ages 40 and 60, belongs to a subject who is a regular yoga practitioner.

Another significant trend, presented in Figure 5, was established by the results, in which the mean detection accuracy of theta stimulus frequencies was detected with higher accuracy in frontal SSVEPs than in occipital SSVEPs, while alpha and beta frequencies were detected with higher accuracy in occipital SSVEPs. Furthermore, detection accuracies in occipital SSVEPs have lower variation levels, as demonstrated by fairly shorter error bars. This suggests that detection accuracy of stimulus frequencies in SSVEPs elicited in the occipital region have greater reliability. These trends can be attributed to the origin of theta and alpha SSVEP signals: While the primary source of theta waves is the frontal midline, alpha waves predominate in the occipital cortex. Thus, one can infer that the detection accuracy of frequencies pertaining to particular bands, found in specific regions in the human brain, depends on the location of the SSVEP being analyzed.

An artificial neural network for predicting cognitive age was trained using detection accuracy of alpha stimulus frequencies in occipital SSVEPs and band power of alpha frequencies in the SSVEP signal, as these were the best indicators of cognitive decline in this study. As shown in Figure 6c, the correlation coefficient between the network outputs and the target outputs is relatively high, showing that the model fits the data well.

The OpenBCI system which was used to collect the EEG data, had high levels of impedance when placing electrodes on the subject's scalp; thus, conducting gel was used to lower the impedance. Furthermore, the data collected from the frontal region had many artifacts compared to data collected from the occipital region, including eye blinks. These were removed by bandpass filtering, but frontal EEG data displayed significantly more error than occipital EEG data. This study can be improved with a broader subject population and sample size; furthermore, in order to achieve a larger level of specificity with this method, it is aimed to test the method on patients with mild cognitive impairment and explore other SSVEP features that can be used as indicators of cognitive deterioration. This study shows that SSVEP-based diagnosis BCI system can be used to verify cognitive deterioration due to aging.

**Supplementary Materials:** The following are available online at http://www.mdpi.com/2504-2289/3/2/29/s1, Table S1: Mean theta amplitudes, Table S2: Mean alpha amplitudes, Table S3: Mean beta amplitudes.

**Author Contributions:** Conceptualization, V.M. and S.S.; methodology, S.S. and V.M.; software, S.S. and V.M.; validation, V.M. and S.S. formal analysis, S.S.; investigation, V.M. and S.S.; resources, S.S.; data curation, S.S.; writing—original draft preparation, S.S.; writing—review and editing, S.S.; visualization, V.M. and S.S.; supervision, V.M.; project administration, S.S.

**Funding:** This research received no external funding.

**Acknowledgments:** We want to thank the following graduate students for their invaluable help during this study: Cesar Aceros (UPRM) and Jennifer Ramirez (UPRM) for their technical assistance during the process of coding and experimentation, and Greg Palmer and Mike Patton (UWisc-Madison), for their feedback and suggestions during data analysis and manuscript preparation. We would also like to thank Mrs. Evelyn Montalvo, for her inspiration, mentorship, feedback, suggestions, and manuscript revisions, which were instrumental to this study.

**Conflicts of Interest:** The authors declare no conflict of interest.

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
