# Peer review of "Assessment of Cognitive Aging Using an SSVEP-Based Brain–Computer Interface System"

_2504-2289, doi:10.3390/bdcc3020029_

Round 1

Reviewer 1 Report

The authors tried to detect cognitive aging using an SSVEP-based BCI. The topic is interesting, but the paper is not written in a professional way. As this stage, a major revision is required to address the following comments.

Although BCI systems usually are referred as online system, this work is done in offline way. It can be considered as a BCI diagnosis system.

1- line 77: Please investigate more in literature survey for relevant papers/citations. There are a bunch of relevant works published for diagnosis of AD using some minutes of recording EEG during open eyes and close eyes. It’s worth briefing these papers in this study. As examples see:

[1]          J. McBride, X. Zhao, N. Munro, C. Smith, G. Jicha, and Y. Jiang, "Resting EEG discrimination of early stage Alzheimer’s disease from normal aging using inter-channel coherence network graphs," Annals of biomedical engineering, vol. 41, pp. 1233-1242, 2013.

[2]          J. C. McBride, X. Zhao, N. B. Munro, C. D. Smith, G. A. Jicha, L. Hively, et al., "Spectral and complexity analysis of scalp EEG characteristics for mild cognitive impairment and early Alzheimer's disease," Computer methods and programs in biomedicine, vol. 114, pp. 153-163, 2014.

2- line 33 and 34: international 10/20 is one of the EEG systems not the only one.

3- paragraph; line 88 to 92: its better to move this paragraph to discussion section at the end as limitation of SSVEP-based BCI.

4- OpenBCI software/hardware is not a good recording EEG system. Please mention your software/hardware limitation in discussion section.

5- The figure 3 shows the EEG is recorded from forehead (frontal??!!). Please discuss the effect of artifacts in the signal processing. Some researchers do not consider these signals as EEG.

6- Line 147: 1 ft (?). All units in a paper must follow ISI standard.

7- Figure 3 is not readable. Please revise all figures and make them consistent for fonts and clarity.

8- Numbering all equations inside the paper and refer them with numbers.

9- ANOVA result should be mentioned in result section rather than discussion section.

10- First paragraph of result section (line 198-201): I can not see the flow of this paragraph in this location and its connection with result section. Start result section with more relevant information.

11- Where is Table 1.

12- Where is Table 5.

13- I ca not see the point for sharing the codes and including them as appendix.

14- Overall, please clarify that you are calculating individual accuracy using SSVEP? Or you are classifying between groups using SSVEP? You are doing calculation inter-group only? How about classification results for intra-group?

Author Response

Reviewer 1:

1) The BCI system has been written as a diagnosis system.  More literature has been added related to aging which is the topic addressed in the paper.

2) International 10//20 system has been corrected as one of the EEG systems

3) Paragraph lines 88 to 92 has been moved to discussion section.

4) The openBCI system limitations has been mentioned in discussion section.

5) The artifacts in EEG recorded from frontal electrodes is mentioned.

6) The units have been corrected.

7) All Figures have been corrected for clarity.

8) All equations have been referred with numbers.

9) ANOVA results have been moved to discussion section.

10) The results section has been corrected.

11) Table 1 is placed correctly.

12) Table 5 is placed correctly.

13) Codes have been removed.

14) Classification for intra-group is presented.

Reviewer 2 Report

In this manuscript the authors used EEG signal, which was recorded during visual stimulus task (Steady-State Visually Evoked Potential) for age discrimination between subjects.

-          The aim of this study is mainly to detect age difference between candidates, and the “cognitive deterioration” is used in the title as well as in the text time to time. On the other hand, you could find the AD term (Alzheimer’s disease) frequently in the text, which I think could be misleading. The proposed method has the potential to be used for differentiation of patients in early stages of AD, although in this study subjects are healthy and it is better to stick more to “cognitive deterioration” term in the text rather than AD except for explaining the potential applications of this study.

For example:

“In terms of BCI use for early diagnosis of Alzheimer’s Disease, or other neurodegenerative disorders for that matter, very few research has been conducted thus far.”

This might cause reader to think this study is about AD detection.

-          Based on my previous comment, I would suggest the authors to include more literature with the same approach they have, which means age discrimination, rather than completely focusing on AD diagnosis.

-          Page 1- Line 31:

Please change ECG to EEG

-          Line 76:

“In terms of BCI use for early diagnosis of Alzheimer’s Disease, or other neurodegenerative disorders for that matter, very few research has been conducted thus far.”

Line 83: 

“Few research is directed towards early diagnostics and treatment of Alzheimer’s Disease through Brain-Computer Interfacing because the cognitive deficits in these patients present challenges to traditional brain-computer interfacing (associated mental activity), and the effects of such disorders on the previously mentioned EEG markers remains largely unexplored[7].”

I don’t agree with the authors. There has been an extensive amount of studies during the past decade for early diagnosis of Alzheimer’s Disease including brain computer interface approaches. Please focus more on your specific methodology to emphasis on the novelty of this study if that is your point.

-          Line 77 to 81

Please mention the reference number as well after “Hedges, et al”, “Hsu, et al”, “Liberati, et al”.

-          The introduction section should be revised in order to make it more concise and on the other hand to add a comprehensive review of the previous works, results, drawbacks,…

-          Line 87:

“Another major limitation of current BCIs is their size and bulkiness; most SSVEP-BCIs require complex setups including LED lights, signal transmission hardware, many cumbersome wires, and EEG caps with approximately 32-180 electrodes.”

please bring some references: “most SSVEP-BCIs require complex setups including LED lights”

I don’t think this is a fair claim. The authors are using the same brain imaging modality (EEG) that they are blaming to be bulky. The only difference here might be less number of EEG electrodes and this cannot get assigned to EEG modality in general.

-          Please refer to figure (1), (2), and (3) in the text.

-          Figure 2 aims to show location of frontal and occipital EEG electrodes. They could remove this figure and instead it would be more helpful and necessary to mention the electrode’s name (e.g Fp1, Fp2, O1, O2, …).

-          Please explain using the headband and how were you able to make sure the location of EEG electrodes is consistent with standard EEG system?

-          Figure 3 should be replaced to have a better resolution to see the details. Especially the back-head set-up is not clear.

-          The purpose of figure 1 is not clear. If the only purpose is to show the size of the flickering pixel you could simply just mention that in the text.

-          In the EEG recording section please mention the sampling rate of 250 Hz.

-          Line 159:

“Canonical correlation analysis (CCA) was used to extract the SSVEP peaks produced at the stimulus frequencies.”

Please bring a reference.

-          Line 190:

“70% of the data (12 samples) were used for training, 15% (2 samples) for validation, and 15% for testing.”

Please make it clear if you have 16 samples/subject or 14 (12+2) samples/subject.

-          The number of total samples that has been used for training and test (12, 2) is insufficient.  It is not clear if the authors extracted their feature from the epochs or not. If yes it is not clear why they have such a low number of samples/subject.

In my experience this number of sample is not sufficient to train any system specially NN.

Author Response

Reviewer 2:

The introduction has been changed to focus on cognitive deterioration and not AD.  More literature has been added with similar approach.

Page 1 line 31:  ECG has been changed to EEG.

Line 76: this line has been removed.

Line 83: this line has been removed.

Line 77 to 81: reference numbers have been added.

The introduction section has been revised to more related literature.

Line 87: The sentences on nature of EEG system is removed, as they are not in general complex.

Figure 2 is removed, and electrode names have been added.

The use of headband and how electrodes are located has been explained.

Figure 3 has better resolution now.

Figure 1 is removed, and the size of the flickering pixel is mentioned. 

The sampling rate of 250 Hz is mentioned in the EEG recording section.

Line 159: Reference to CCA has been added.

Line 190: Number of samples per subject is added.

The correct number of samples used for training the NN is mentioned.

Thank you!

Round 2

Reviewer 1 Report

Figures 2 and 3 need to be improved for better quality.

Caption in Figure 3 is missing.

There are (a), (b), (c) inside plots of Figures 6 and 7. Please revise them.

Table 5 is broken into two pages. It is hard to follow. Please correct that.

Author Response

Thank you for the feedback. All of the comments were addressed and minor edits were made to the manuscript as needed.

The size and resolution of Figures 2 and 3 were increased to improve readability.

The caption of Figure 3 was hidden by the figure; this was adjusted and the caption is now visible.

(a), (b), and (c) was moved to their appropriate locations in their corresponding figures.

Table 5 was edited so that it falls on only one page.